# Digital Staining of Unpaired White and Blue Light Cystoscopy Videos for Bladder Cancer Detection in the Clinic

**Shuang Chang**[1]                                SHUANG.CHANG@VANDERBILT.EDU
**Haoli Yin**[2]                                     HAOLI.YIN@VANDERBILT.EDU
**Kristen Scarpato**[3]                            KRISTEN.R.SCARPATO@VUMC.ORG
**Amy Luckenbaugh**[3]                         AMY.N.LUCKENBAUGH@VUMC.ORG
**Sam Chang**[3]                                       SAM.CHANG@VUMC.ORG
**Christian Bolenz**[4]                              CHRISTIAN.BOLENZ@UMM.DE
**Maximilian C. Kriegmair**[5,6]       MAXIMILIAN.KRIEGMAIR@MEDMA.UNI-HEIDELBERG.DE
**Nikolaos C. Deliolanis**[6]                      N.DELIOLANIS@THERICON.COM
**Soheil Kolouri**[2]                               SOHEIL.KOLOURI@VANDERBILT.EDU
**Audrey Bowden**[1,7]                           A.BOWDEN@VANDERBILT.EDU

[1] *Vanderbilt University, Department of Biomedical Engineering, Nashville, TN, United States, 37232*

[2] *Vanderbilt University, Department of Computer Science, Nashville, TN, United States, 37232*

[3] *Vanderbilt University Medical Center, Department of Urology, Nashville, TN, United States, 37232*

[4] *University of Ulm, Department of Urology, Ulm, Germany*

[5] *Urology Hospital Planegg, Department of Urology, Planegg, Germany*

[6] *Heidelberg University, Medical Faculty Mannheim, Mannheim, Germany*

[7] *Vanderbilt University, Department of Electrical Engineering, Nashville, TN, United States, 37232*

**Editors:** Accepted at MIDL 2023

## Abstract

Blue light cystoscopy (BLC) has been shown to detect bladder tumors with better sensitivity than white light cystoscopy (WLC); however, its increased cost and dye administration time have challenged widespread adoption of the technology. Here, we demonstrate a low-cost strategy to generate BLC images directly from WLC images. We performed digital staining of WLC images obtained from tumor resection procedures and demonstrate that the resulting digitally generated BLC images show strong resemblance to ground truth BLC images, with negligible degradation of the image quality.

**Keywords:** Bladder cancer, cystoscopy, style transfer, deep learning.

## 1. Introduction

The high recurrence rate of urothelial carcinoma necessitates repeated surveillance cystoscopy to detect suspicious lesions in the bladder. Left untreated, undetected or incompletely resected non-invasive cancers may progress to the muscle-invasive stage and require aggressive treatment, including removal of the bladder. White Light Cystoscopy (WLC) is commonly used to examine the bladder for suspicious lesions during a transurethral resection of bladder tumor (TURBT) procedure. Blue Light Cystoscopy (BLC) utilizes an exogenous contrasting dye that selectively accumulates in cancerous tissues. With the added contrast,

BLC successfully reduces short-term recurrence by 10% and increases the detection rate of high-grade tumors by 43%, compared to WLC. Despite the sensitivity advantage of BLC, the high cost of the system and the time and space needed to administer the dye prior to imaging have limited availability of BLC to few (less than 5%) hospitals in the U.S. and limited use to the operating room. Moreover, a significant number of patients are unable to retain the dye for the required instillation time. A simple, quick and low-cost strategy to produce BLC images would make improved detection sensitivity accessible to more hospitals and enable affordability of BLC technology for use in clinics.

To address the sensitivity limitation of WLC, a classification-based approach to identify tumor present in WLC frames, CystoNet, was introduced (Shkolyar et al., 2019). However, the model was trained on manually labeled WLC frames, which by definition involves a subset of tumors already detectable by human eyes with white light. To overcome the challenge of low sensitivity, it is important to visualize tumors that are not currently seen under white light imaging. In 2021, Ali et al. introduced a BLC-image-based artificial intelligence diagnostic platform, where they showed the classification of malignant lesions with 95.77% sensitivity (Ali et al., 2021). However, the proposed platform can only be utilized in the few hospitals and clinics where BLC systems are already available.

In our study, we aim to enable dye-free bladder tumor detection by using deep learning to create BLC-like images (that is, digitally generated BLC images) from WLC images that having been digitally stained (Chang et al., 2023). To our knowledge, this is the first demonstration of digital staining on cystoscopy data. Our proposed workflow has the potential to reduce the current gap in bladder cancer detection by improving the detection sensitivity of WLC while increasing the accessibility of BLC-like images without the burdens of cost and dye administration.

## 2. Methods and Results

**Data collection and preparation.** Data used in this study were originally collected for a proof-of-concept study of multiparametric cystoscopy for bladder cancer imaging (Kriegmair et al., 2020). A color camera equipped with a multi-bandpass filter and a multi-LED light source were used to collect near-simultaneous reflectance (i.e., WLC) and fluorescence (i.e., BLC) frames through temporal multiplexing. Near-perfectly registered WLC and BLC videos at a frame rate of 20 Hz were derived from the multispectral data collected, among others; the blue light videos provide ground truth data to evaluate our network. Videos from three patients were used for our study, where the frames included papillary tumors, flat tumors, and normal bladder tissue regions. The videos were concatenated, and paired frames were extracted and cropped to 256 by 256 pixels to create sequential image data. The sequential data were then split to have the first 90% reserved for training and the last 10% reserved for testing as the holdout set. To prepare the training data for the model, the WLC and BLC pairs were first synthetically unpaired by randomizing the order of the BLC frames while keeping the WLC frame order fixed. Then, the frames from both BLC and WLC were randomly split into 80/20 training and validation sets.

**Transfer model.** To create a robust model for semantically-aware modality transfer, we trained our model using unpaired WLC/BLC image data following the Density Changing Regularized Unpaired Image Translation (DECENT) method (Xie et al., 2022). We

employed autoregressive flows for density estimation and used a ResNet-based generator with a PatchGAN discriminator. During the training process, we first updated the density estimators, followed by updating the discriminator and optimizing the generator with the Polyak-averaged version of the density estimator and the LSGAN (Least Squares GAN) objective to help stabilize the learning process. The model consisted of three terms from the original method: an adversarial loss, an identity mapping loss, and a density changing loss. The outputs are the digitally stained WLC frames or, equivalently, dgBLC frames.

**Evaluation metrics.** We defined three categories of analysis metrics that evaluate the staining accuracy, color consistency and overall image quality. Staining accuracy assessment was performed by creating a fluorescence segmentation mask. Using the BLC data as the ground truth, we compared its masks with those for the corresponding dgBLC data and computed the percentages of correctly and incorrectly stained pixels (i.e., red pixel in the ground truth showing up as blue pixel in dgBLC, or vice versa). To assess the realistic appearance of the network output, the color of the dgBLC images was analyzed in the YCbCr color space for each color channel. For overall image quality, both reference-based (FSIM, PSNR) and reference-less (BRISQUE) image quality metrics were computed.

**Table 1** reports the mean and standard deviation of the assessment metrics, where we observed excellent agreement in staining area and color and negligible degradation in overall image quality. **Figure 1** shows two examples of original WLC-BLC pairs and the output dgBLC images. It is important to note that while this proof-of-concept study uses a registered dataset for quality evaluation purposes, our approach does not rely on registered datasets. In the future work, we will train the model using clinically acquired WLC and BLC videos, where the video frames are no longer perfectly registered.

Table 1: Assessment metrics calculated from digitally stained frames from the testing set, using the corresponding BLC frames as reference.

| | Staining accuracy | | | Color consistency | | | Image quality | | | |
|---|---|---|---|---|---|---|---|---|---|---|
| | Correct staining % | Incorrect (r → b) % | Incorrect (b → r) % | Δy | ΔCr | ΔCb | BRISQUE (GT) | BRISQUE (dgBLC) | FSIM | PSNR |
| Mean | 87.98 | 4.406 | 7.659 | 0.02548 | 0.01983 | 0.01161 | 35.41 | 41.07 | 0.9218 | 27.31 |
| Std. Deviation | 4.694 | 1.553 | 3.691 | 0.01054 | 0.006666 | 0.001839 | 4.067 | 3.238 | 0.02864 | 3.397 |

WLC      BLC      dgBLC

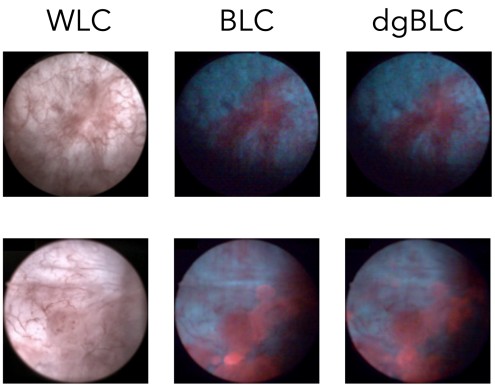

Figure 1: Two examples of WLC-BLC image pair and dgBLC results.

## Acknowledgments

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
