# OpenReview forum: "Digital Staining of Unpaired White and Blue Light Cystoscopy Videos for Bladder Cancer Detection in the Clinic"
_MIDL.io/2023/Short_Paper_Track — MIDL 2023 Short paper track Poster_

### Official Review · Reviewer_FbNK · 2023-04-24

**Rating:** 6
**Confidence:** 4

**Review:**

The work investigates the effectiveness of deep-learning based image translation for generating synthetic Blue Light Cystoscopy (BLC) from White Light Cystoscopy (WLC), in the case of bladder cancer imaging. The motivation is that BLC has been previously shown to increase sensitivity in detecting tumors, in comparison to WLC, but is more expensive and less convenient than WLC, and is therefore much less used / accessible in practice. Therefore, the work seeks whether a GAN network can generate synthetic BLC, given a real WLC. The work is performed using a database of paired WLC and BLC images (and therefore there is real ground truth for evaluating synthetic BLC against real BLC). The work trains a recent variation of a GAN (Xie et al 2022), and evaluates it using multiple metrics. The evaluation shows that promising results.

Strengths:
- The application is interesting, and if synthetic BLC would be proven useful, it could help improve cancer detection.
- The paper is well written.
- The evaluation employes multiple metrics, in an attempt to evaluate different aspects of the synthetic data.

Weaknesses:
- It is unclear whether the fundamental assumption of the work holds. Specifically, BLC's contrasting dye has the property that it "selectively accumulates in cancerous tissues" (as per paper), which in turn makes them visible in BLC. In other words, (real) BLC by construction may be able to show cancers that are not visible in WLC? So, trying to create synthetic BLC from WLC is like asking to generate information (cancer areas) that are not visible in the original image (WLC). This can in turn result in the common problem of image-translation, where a GAN hallucinates information, to make the resulting image to look realistic. On the other hand, perhaps the information is there in the WLC and the blue stain just makes it more visible? I am not sure as I am not an expert in this area. Regardless, the application seems under-explored to me, so it should still be interesting to explore the capabilities of the technology in this area.
- The evaluation is limited. Specifically, it is hard to understand whether the shown results (numbers)  are good or not for the specific application. This is because only 1 method is evaluated, the metrics are presented, but there is nothing to compare those numbers to see if they are "good". There is no baseline. Nor is there an attempt to show whether the images are concidered "good" (e.g. by human raters) or "useful" (is 89% correct staining sufficient for downstream tasks, diagnosis and therapy? Or too low? No insights)
- There is no technical contribution. Only one method is employed (Xia 2022) and evaluated.

Overall, the weaknesses do not allow strong conclusions to be made, eventhough the application is interesting.

---

### Official Review · Reviewer_PLFN · 2023-04-26
**Generating blue light cystoscopy (BLC)  to detect bladder tumors with better sensitivity given only lower cost white light cystoscopy (WLC)**

**Rating:** 5
**Confidence:** 4

**Review:**

This study aims to solve an important problem of generating blue light cystoscopy (BLC)
 to detect bladder tumors with better sensitivity given only white light cystoscopy (WLC) which is less costly to obtain and more widely available. The authors used the Density Changing Regularized Unpaired Image Translation (DECENT) method (Xie et al., 2022) for modality transfer of WLC to BLC. Previous work showed that the BLC-image-based artificial intelligence diagnostic platform could classify malignant lesions with  95.77% sensitivity. And the goal of this work seems to be to achieve this with generated BLC images.  Unfortunately the results show only training and testing on 3 subjects where part of the sequential video data from the same subjects were used for training and rest for testing, which is not an acceptable method. The resultant metrics reported are also hard to interpret as there is no metric reported for whether the resultant images can successfully be used to classify malignant lesions.